# Spatially Resolved Proteomic and Transcriptomic Profiling of Anaplastic Lymphoma Kinase-Rearranged Pulmonary Adenocarcinomas Reveals Key Players in Inter- and Intratumoral Heterogeneity

**DOI:** 10.3390/ijms241411369

**Published:** 2023-07-12

**Authors:** Beáta Szeitz, Tibor Glasz, Zoltán Herold, Gábor Tóth, Mirjam Balbisi, János Fillinger, Szabolcs Horváth, Réka Mohácsi, Ho Jeong Kwon, Judit Moldvay, Lilla Turiák, Attila Marcell Szász

**Affiliations:** 1Division of Oncology, Department of Internal Medicine and Oncology, Semmelweis University, 1083 Budapest, Hungary; szeitz.beata@phd.semmelweis.hu (B.S.);; 2Department of Pathology, Forensic and Insurance Medicine, Semmelweis University, 1091 Budapest, Hungary; 3MS Proteomics Research Group, Institute of Organic Chemistry, Research Centre for Natural Sciences, 1117 Budapest, Hungary; 4Doctoral School of Pharmaceutical Sciences, Semmelweis University, 1085 Budapest, Hungary; 5Department of Pathology, National Korányi Institute of Pulmonology, 1121 Budapest, Hungary; 6Department of Biotechnology, Division of Life Sciences, Yonsei University, Seoul 03722, Republic of Korea; 71st Department of Pulmonology, National Korányi Institute of Pulmonology, 1121 Budapest, Hungary; 8Department of Tumor Biology, National Korányi Institute of Pulmonology, 1121 Budapest, Hungary; 9Department of Bioinformatics, Semmelweis University, 1094 Budapest, Hungary

**Keywords:** lung adenocarcinoma, pulmonary adenocarcinoma, *ALK* rearrangement, multi-omics, digital spatial profiling, proteomics, transcriptomics

## Abstract

Pulmonary adenocarcinomas (pADCs) with an *ALK* rearrangement are a rare cancer subtype, necessitating comprehensive molecular investigations to unravel their heterogeneity and improve therapeutic strategies. In this pilot study, we employed spatial transcriptomic (NanoString GeoMx) and proteomic profiling to investigate seven treatment-naïve pADCs with an *ALK* rearrangement. On each FFPE tumor slide, 12 smaller and 2–6 larger histopathologically annotated regions were selected for transcriptomic and proteomic analysis, respectively. The correlation between proteomics and transcriptomics was modest (average Pearson’s *r* = 0.43 at the gene level). Intertumoral heterogeneity was more pronounced than intratumoral heterogeneity, and normal adjacent tissue exhibited distinct molecular characteristics. We identified potential markers and dysregulated pathways associated with tumors, with a varying extent of immune infiltration, as well as with mucin and stroma content. Notably, some markers appeared to be specific to the ALK-driven subset of pADCs. Our data showed that within tumors, elements of the extracellular matrix, including *FN1*, exhibited substantial variability. Additionally, we mapped the co-localization patterns of tumor microenvironment elements. This study represents the first spatially resolved profiling of ALK-driven pADCs at both the gene and protein expression levels. Our findings may contribute to a better understanding of this cancer type prior to treatment with ALK inhibitors.

## 1. Introduction

Lung cancer is the leading cause of cancer mortality worldwide, responsible for 2.21 million out of nearly 10 million cancer deaths globally in 2020 [1]. The most common histological type of non-small cell lung cancer (NSCLC) is pulmonary adenocarcinoma (pADC), with the morphological subtypes including acinar, lepidic, micropapillary, papillary and solid patterns [2]. The acinar (or tubular) subtype is characterized by the formation of glandular structures with luminal spaces. The papillary subtype is distinguished by the formation of papillary structures, which are finger-like projections lined by tumor cells. The solid subtype, on the other hand, is composed of sheets or nests of tumor cells without any glandular or papillary structures. Importantly, these histopathological characteristics also reflect on the tumor behavior, such as therapy resistance, and can impact patient outcomes [3]. In addition to morphological subgroups, pADCs can further be classified into several molecular subtypes with distinct characteristics that reflect on clinicopathological data as well [4,5,6,7,8]. The proposed classification systems are crucial in understanding the heterogeneity of lung cancer and guiding personalized treatment approaches.

While most pADCs lack an identifiable driver oncogene, 3–5% of all pADCs within the Caucasian population contain anaplastic lymphoma kinase (*ALK*) gene rearrangements [9]. After *ALK* fuses with its partner gene (most frequently *EML4*), an increase in ALK activity occurs, which activates downstream signaling cascades such as the MAPK, PI3K–AKT or JAK–STAT pathways [10]. Since the discovery of *ALK*-rearranged lung cancers in 2007 [11], several ALK inhibitors have been developed and approved for clinical use, such as the first-generation ALK inhibitor Crizotinib (Xalkori), or the second-generation inhibitor Alectinib (Alecensa) [12]. These drugs significantly improve progression-free survival and overall response rates in *ALK*-positive pADCs; however, responses to ALK inhibitors are often not durable, and tumors acquire resistance to treatment [12]. Importantly, intratumoral heterogeneity has been associated with poor patient outcomes and therapeutic resistance in many different cancers [13]. The presence of tumor cells with heterogeneous phenotypes within the same tumor is a complex phenomenon that stems from various genetic, epigenetic and environmental inputs, and is further complicated by the complex interactions between cancer cells and the tumor microenvironment (TME) [13]. The TME consists of immune cells (B cells, natural killer cells, T cells and tumor-associated macrophages), as well as adipocytes, endothelial cells, fibroblasts and mesenchymal stem cells [14]. In addition, the TME itself is a heterogeneous entity where the crosstalk among its various elements also influences cancer-related processes [14]. 

So far, large-scale omics studies in pADC, containing only a handful of ALK-driven tumors, have utilized bulk expression profiling [4,5,6,7,8], failing to capture intratumoral molecular changes that carry important implications for therapy and patient survival. Spatially resolved transcriptomic or proteomic studies of NSCLCs have been already utilized for biomarker discovery, in particular contributing to potential improvements in immunotherapy approaches [15,16,17,18]. Recently, we reported, for the first time, the proteomic and glycosaminoglycan characterization of seven pADCs with *ALK* rearrangement in a spatially resolved manner [19]. As demonstrated by a multitude of lung cancer studies [5,6,7,8], the integration of multiple omic datasets, such as genomics, transcriptomics or proteomics, can capture the complexity of the disease more effectively. Herein, we extended our previous proteomic results [19] with NanoString GeoMx gene expression profiling, enabling a spatial multi-omic characterization of ALK-driven pADCs prior to treatment with ALK inhibitors. First, we mapped protein and gene expression patterns and dysregulated pathways characteristic to tumor regions with distinct histopathological features. Next, we investigated the intratumoral heterogeneity detectable at both molecular levels, demonstrating the key role of extracellular matrix elements in this phenomenon. By studying inter- and intratumoral differences, this pilot study aims to gain further insight into the molecular landscape of this cancer type.

## 2. Results

### 2.1. Spatially Resolved Characterization of the Proteome and Transcriptome in ALK-Rearranged pADCs

To characterize inter- and intratumoral heterogeneity in *ALK*-rearranged pADCs, we assembled a cohort of seven formalin-fixed, paraffin-embedded (FFPE), treatment-naive primary tumors with confirmed *ALK* rearrangements (Table 1). The cohort consisted of three female and four male patients. The mean age at surgery was 57.0 years (standard deviation, SD = 14.1). Five and two patients received Crizotinib (Xalkori) and Alectinib (Alecensa) after sample collection, respectively. The mean overall survival was 6.5 years (SD = 3.6). By the censoring date (January/February 2022), two patients died of lung cancer, both of whom received Crizotinib.

#### 2.1.1. Histopathological Description of the Sample Cohort

For each FFPE tumor, smaller regions of interest (ROIs) were selected for transcriptome and proteome analysis (tROIs and pROIs, respectively). The smaller tROIs were chosen based on their morphologic setting and varying levels of lymphocytic infiltration, while the pROIs covered larger areas of the whole tumor slide. Histologically normal adjacent lung tissue near the tumor (normal adjacent tissue, NAT) was also obtained for both proteomic and transcriptomic analysis. In total, we analyzed 84 tROIs and 23 pROIs (Appendix A, Appendix A). An overview of the study can be seen in Figure 1a.

Each ROI was annotated with histopathological data (see Table 1 for the overview and Appendix A and Appendix A for the detailed ROI characteristics, also including the relationship between pROIs and tROIs). For the pROIs, the percentage of tumor-infiltrating lymphocytes (TILs), the mucin and stroma score (ranging between 0 and 3) were recorded, whereas for the tROIs, the immune score (ranging between 0 and 3) was noted (Section 4). Additionally, the morphology of both the pROIs and tROIs from tumor regions was described (tubular, papillary or solid).

#### 2.1.2. Overview of the Collected Proteomic and Transcriptomic Data

Through label-free proteomics and NanoString GeoMx Digital Spatial Profiling (Section 4, Appendix A), we identified and quantified a total of 2318 proteins and 1811 genes. While the transcriptomic data contained no missing values, the proteomic data showed a substantial amount of missing protein intensities, with only 49.8% of the protein groups (1154 out of 2318) being quantified across at least 80% of the analyzed pROIs. For statistical analyses, the missing values for the 1154 protein groups were imputed with low numbers, assuming that the missingness was due to low abundance [20]. The number of imputed values was the highest for the NAT regions (Appendix A). A visual inspection of the principal component analysis (PCA) biplot based on the Z-score-normalized gene counts (Appendix A) and label-free quantification (LFQ) values (Appendix A) showed no outlier ROIs in either dataset. Importantly, the imputation did not alter the overall clustering of the proteomic samples (Appendix A).

The filtered protein list from the pROIs and the gene list from the tROIs shared merely 162 common genes. To circumvent this low overlap at the gene level, both datasets were additionally transformed into single-sample gene set scores (singscores) (Section 4). This increased the overlap to 431 common singscores. The correlation between the two omics data was assessed by averaging the gene count data and tROI singscores across the larger measured pROIs. The commonly identified genes (*n* = 162) were generally positively correlated between the two datasets (Figure 1b, Appendix A), with a mean Pearson correlation coefficient (*r*) of 0.43 (SD = 0.33). To be exact, 69 of the 162 genes showed a strong positive Pearson correlation (adj. *p* < 0.05) and only 1 gene (60S acidic ribosomal protein P0, *RPLP0*) showed a significant negative Pearson correlation (adj. *p* < 0.05). The top positively correlating genes were glucose-6-phosphate 1-dehydrogenase (*G6PD*), 14-3-3 protein sigma (*SFN*), fructose-bisphosphate aldolase A (*ALDOA*), thymidine phosphorylase (*TYMP*) and carcinoembryonic antigen-related cell adhesion molecule 6 (*CEACAM6*) (Appendix A). The functional analysis of the 69 significantly positively correlating genes revealed the overrepresentation (one-tailed Fisher’s exact test, *p* < 0.15) of the following gene sets: mTORC1 signaling, glycolysis and gluconeogenesis, G2/M checkpoint, transcriptional regulation by *TP53*, tyrosine and phenylalanine metabolism, and allograft rejection (Figure 1b, Appendix A).

On the other hand, the correlation between tROI and pROI singscores (*n* = 431) were less prominent (Figure 1c, Appendix A), with a mean Pearson’s *r* of 0.24 (SD = 0.34). In total, 63 and 6 gene sets showed a significant positive and negative correlation, respectively, from which the top 5 significant processes in both directions are indicated in Figure 1c. Among the top positively correlating processes, we identified metabolism (e.g., glycolysis, RNA metabolism) or immune-system-related processes (such as allograft rejection, adaptive immune system), as well as cell cycle and DNA replication, signal transduction and gene regulation (mTORC1 signaling, MYC targets, p53 pathway, NCAM signaling) (Figure 1c, Appendix A). The negatively correlating processes, with adj. *p* < 0.05, included oncogenic MAPK signaling, MAP2K and MAPK activation, and developmental biology (Figure 1c, Appendix A).

### 2.2. Multi-Omic Signatures of Histopathological Features

To uncover the main ROI characteristics affecting the protein and gene expression profiles, we performed the unsupervised consensus clustering (partitioning around medoids algorithm, Pearson distance) of the pROIs and tROIs (Figure 2a,b). This exploratory analysis revealed that both sample sets were optimally grouped into six clusters and clustering was largely driven by interpatient differences. In addition, NAT regions were well separated based on their proteomic profile, which was less prominent at the gene expression level.

Next, we were interested in how various histopathological features, including tissue type, immune infiltration, mucin and stroma score were reflected on the proteome and transcriptome. Therefore, differential expression analyses, followed by pre-ranked gene set enrichment analysis (GSEA) were conducted. In parallel, we also investigated which protein/gene-level finding could be supported by the Clinical Proteomic Tumor Analysis Consortium (CPTAC) [6] pADC data.

#### 2.2.1. Molecular Characteristics Associated with pADCs Compared to Normal Adjacent Tissues

The pROIs showed prominent differences between tumors and NATs, with 310 proteins upregulated and 136 proteins downregulated in tumors (Appendix A). At the tROI level, 47 upregulated and 38 downregulated genes in tumors were detected (Appendix A). In both datasets, *ALDOA*, glutathione peroxidase 1 (*GPX1*), macrophage migration inhibitory factor (*MIF*), pyruvate kinase PKM (*PKM*), endoplasmin (*HSP90B1*), mucin-1 (*MUC1*), tenascin (*TNC*), *SFN*, *CEACAM6* and 40S ribosomal protein S6 (*RPS6*) were significantly upregulated in tumors (Appendix A), whereas the collagen alpha-6(VI) chain (*COL6A6*), alpha-2-macroglobulin (*A2M*) and plasma protease C1 inhibitor (*SERPING1*) were significantly downregulated in tumors (Appendix A). The genes complement C3 (*C3*), C4b-binding protein alpha chain (*C4BPA*), clusterin (*CLU*), collagen alpha-1(I) chain (*COL1A1*) and laminin subunit beta-3 (*LAMB3*) showed, however, opposite tendencies in the two omic data (downregulation in tumors at the pROI level, but upregulation in tumors at the tROI level) (Appendix A). The CPTAC dataset largely supported our findings for these proteins and genes, either at the proteome or transcriptome level, or both (Appendix A). Interestingly, some differential expressions (*GPX1*, *C4BPA*) were only confirmed in the *ALK*-rearranged subset of the CPTAC dataset.

Pathway-level analysis of tumor-specific signatures, supported by both pROI and tROI data, showed that tumors display an increased expression of members of glycolysis, unfolded protein response, mTORC1 signaling and infection pathways (Figure 2c, Appendix A). Moreover, proteomics indicated that proteins involved in pathways such as translation, RNA and glucose and amino acid metabolisms were upregulated in tumors compared to NATs. On the other hand, numerous processes related to the extracellular matrix (ECM) organization, apical junction, receptor tyrosine kinase (RTK) signaling such as MAPK signaling and Toll-like receptors were significantly downregulated in tumor pROIs. The complement and coagulation cascade displayed opposite tendencies at the two molecular layers (upregulated according to the tROI data, but downregulated according to the pROI data).

#### 2.2.2. Multi-Omic Signatures Related to Varying Levels of Immune Infiltration

In terms of immune-infiltration-associated differences, only the protein sulfotransferase 1A1 (*SULT1A1*) was significantly upregulated with increasing TIL amounts in the pROIs, while tROI data showed the upregulation of HLA class I histocompatibility antigen, B alpha chain (*HLA-B*) and the downregulation of prostaglandin G/H synthase 2 (*PTGS2*), with an increasing immune score (Appendix A, Appendix A). By correlating these genes’ expression values in the CPTAC data with the ESTIMATE (estimation of stromal and immune cells in malignant tumor tissues using expression data [21]) immune scores, our observations could be confirmed, either at the protein or transcript level or at both levels (Appendix A). Interestingly, *SULT1A1* only showed upregulation in the *ALK*-rearranged pADCs at the transcript level.

The observation that proteomic regions did not significantly differ based on TIL amounts was underscored by the pre-ranked GSEA results (no gene set below adj. *p* < 0.05), whereas tROIs showed prominent differences based on the immune score, and most pathways upregulated with a higher immune score were involved in the immune system, such as antigen processing and presentation, signaling by interleukins, or neutrophil degranulation (Figure 2d, Appendix A).

#### 2.2.3. Proteomic Changes Associated with Mucin and Stroma Scores

The mucin- and stroma-score-related differences were only assessed for pROIs. The proteins prothrombin (*F2*), lumican (*LUM*), prolargin (*PRELP*) and N-acetylmuramoyl-L-alanine amidase (*PGLYRP2*) were significantly upregulated, while 13 proteins such as lactotransferrin (*LTF*), proteins S100-A8 and -A9 (*S100A8*, *S100A9*), histone H1.5 (*HIST1H1B*), DNA-dependent protein kinase catalytic subunit (*PRKDC*), or splicing factor U2AF 65 kDa subunit (*U2AF2*) were significantly downregulated with increasing mucin score (Appendix A). On the other hand, the protein biglycan (*BGN*) was significantly upregulated, and voltage-dependent, anion-selective channel protein 2 (*VDAC2*), long-chain-fatty-acid–CoA ligase 1 (*ACSL1*) and hemoglobin subunit delta (*HBD*) were significantly downregulated with increasing stroma scores (Appendix A).

To investigate the alignment with CPTAC data, we compared the above-mentioned protein and gene expressions in tumors with invasive mucinous morphology (*n* = 3) to tumors with other morphologies (acinar, intestinal, lepidic, micropapillary, papillary, sarcomatoid, solid). Only a subset of the differentially expressed proteins could be confirmed by the CPTAC data, namely *HIST1H1B*, *PRKDC*, *U2AF2* and *PRELP* (Appendix A). In addition, for proteins showing differential expression with stroma score in our data, we checked the correlation between ESTIMATE stromal scores and the protein or gene expression in the CPTAC data. Only the trend for *VDAC2* and *ACSL1* could not be validated (Appendix A). Of note, the proteins showing an increasing and decreasing expression tendency with mucin and stroma scores were not overlapping (Appendix A); however, the pre-ranked GSEA results showed similar patterns for both mucin and stroma-high ROIs, from which the most prominent was the concordant upregulation of ECM organization, epithelial–mesenchymal transition (EMT)-related proteins, and in parallel, the downregulation of members of RNA metabolism or signaling by ROBO receptors (Figure 2e).

### 2.3. Assessment of Intratumoral Heterogeneity in Seven ALK-Rearranged pADC Cases

Our previous analyses were focused on the multi-omics characterization of histologically defined ROI groups. However, intratumoral heterogeneity might not be solely defined by visible histopathological observations. To uncover the drivers of molecular homogeneity and heterogeneity, the overall expression variability for each protein and gene within each case was assessed through their coefficient of variation (CV) (Section 4).

#### 2.3.1. Homogeneously Expressed Proteins and Genes within the Tumors and Associated Pathways

Proteins with a stable expression (i.e., had a low CV) within minimum four tumors (Section 4) (*n* = 49) showed marginally significant (one-sided Fisher’s exact test, adj. *p* < 0.25) enrichment for signaling processes such as PI3K–AKT–mTOR signaling, EPH–ephrin signaling, oncogenic MAPK signaling and signaling by Rho GTPases (Figure 3a, Appendix A). The tROI data showed that genes with a stable expression within minimum four tumors (*n* = 276) were involved in pathways such as chromatin organization, DNA repair, transcription, post-translational protein modification (e.g., ubiquitination, SUMOylation), innate immune system (MyD88-independent TLR4 cascade, DDX58/IFIH1-mediated induction of interferon-alpha/beta, Fc epsilon receptor signaling), death receptor signaling, just to highlight the top significant gene sets (Appendix A). Notably, oncogenic MAPK signaling and PI3K–AKT–mTOR signaling were also overrepresented (one-sided Fisher’s exact test, adj. *p* = 0.011 and 0.115) at the tROI level (Figure 3a).

#### 2.3.2. Key Players in Intratumoral Heterogeneity

To study the molecular heterogeneity of the tumors, the proteins and genes that appeared as highly variable for at least four cases (*n* = 101 and 232, respectively) were investigated (Appendix A). Notably, three heat shock proteins, namely heat shock protein family A (Hsp70) member 1A (*HSPA1A*), heat shock protein family B (Small) member 1 (*HSPB1*) and heat shock protein 90 beta family member 1 (*HSP90B1*) displayed high stability at the protein expression level, but high variability at the gene expression level in at least four tumors (Appendix A).

The pathway enrichment analysis for the variable proteins and genes (Figure 3a, Appendix A) revealed that multiple members of the ECM organization and remodeling, ECM–cell interactions, EMT, oxygen transport and specific pathways “MET activates PTK2 signaling”, “diseases associated with glycosaminoglycan metabolism”, “molecules associated with elastic fibers” showed high variability across the pROIs. For tROIs, members of EMT, complement and coagulation cascades, ECM organization, signaling pathways (TNFa signaling via NFkB, KRAS signaling, MET-activated PTK2 signaling), cell motility and migration, receptor interactions, angiogenesis and hypoxia, glycolysis and inflammatory responses appeared to be highly variable.

Among all six tumors that contained multiple tumor pROIs, we identified five proteins with a high CV (i.e., were among the proteins with the top 20% highest CV in all tumors), namely spectrin alpha chain, erythrocytic 1 (*SPTA1*), apolipoprotein E (*APOE*), band 3 anion transport protein (*SLC4A1*), fibronectin (*FN1*) and periostin (*POSTN*) (Appendix A). In line with the pathway analysis results, all genes except for *SPTA1* were members of at least one of the significant pathways (Appendix A).

We detected 24 genes across the tumor tROIs that showed a high CV (i.e., were among the genes with the top 20% highest CV in seven out of seven cases), such as multiple collagens (*COL1A1*, *COL1A2*, *COL3A1*, *COL5A1*, *COL5A2*), *S100A8*, *S100A9*, *FN1*, or Early growth response protein 1 (*EGR1*) (Appendix A). The majority of genes were also members of the significantly enriched pathways (Appendix A).

Of note, *FN1* showed high variability at both molecular layers, which can be demonstrated via sketch images of all seven tumors (Figure 3b). *FN1* levels were mildly correlated across the tumor pROIs and tROIs (Pearson’s *r* = 0.3548, *p* = 0.1485, adj. *p* = 0.2292, Appendix A). In addition, *FN1* was associated with 12 out of the 15 gene sets enriched at both molecular levels for variably expressed proteins and genes (Appendix A), including EMT, which was among the top significant pathways in both pROI and tROI data. The EMT singscores strongly correlated with the *FN1* amounts across both pROIs and tROIs (Figure 3c). Hypothesizing that heterogeneous *FN1* expression could have implications for patient outcome, we examined its relationship to the overall survival across multiple pADC cohorts, including the cohort in the present study, as well as the pADC cohort of The Cancer Genome Atlas (TCGA) and CPTAC. We found that *FN1* could not be significantly associated with survival in a univariate setting in neither cohorts, albeit the hazard ratios were generally higher than 1, particularly for the *ALK*-rearranged subset of pADCs (Figure 3d).

#### 2.3.3. Co-Localization Patterns of Tumor Microenvironment Elements

To further investigate the intratumoral heterogeneity, we performed the estimation of immune and stroma cell abundance across tROIs to reflect on their TME composition. The unsupervised k-means clustering on the TME elements showed interesting co-localization patterns (Figure 3e). In particular, macrophages, non-classical monocytes, endothelial cells, natural killer cells and fibroblasts formed one cluster (row cluster 1). Row cluster 2, rather similar to row cluster 1, contained naive and memory CD4+ T cells, naive and memory B cells, memory CD8+ T cells, regulatory T cells, plasma and plasmacytoid dendritic cells (DCs). Interestingly, the abundance of naive CD8+ T cells, mast cells, myeloid DCs, neutrophil cells and classical monocytes (row cluster 3) were not correlated with other TME elements, only weakly correlated with each other (Appendix A).

## 3. Discussion

Spatial molecular profiling of tumors continues to enhance our understanding of cancer by adding to the crucial spatial context, hence the increasing popularity of such studies, also in lung cancer [15,16,17,18]. Harnessing the proteome for spatial characterization is particularly valuable alongside DNA and RNA sequencing, as ultimately, proteins provide the structural and functional framework for cellular life and thus offer a closer representation of the phenotype. In this preliminary study, we explored the main molecular and pathway-level drivers of inter- and intratumoral heterogeneity across seven pADCs with confirmed *ALK* rearrangements, collected prior to treatment with ALK inhibitors (either Crizotinib or Alectinib).

On the FFPE slides, larger proteomic and smaller transcriptomic regions (23 pROIs and 84 tROIs), both from tumor and NAT regions, were selected for molecular profiling. Regions were characterized according to histopathological observations, including morphology and the extent of immune infiltration, as well as mucin and stroma scores. Morphological patterns (lepidic, papillary, acinar, cribriform, micropapillary and solid) and further histological features are often combined within a pADC and carry prognostic value [3,22]. In our study, only one tumor (Case 3) displayed multiple morphologies, and a multitude of regions (10 out of 23 pROIs and 44 out of 84 tROIs) showed solid patterns. Tumors with a solid morphology were generally characterized by a low mucin score compared to papillary and tubular morphologies.

Through molecular profiling, we achieved the identification of 2318 protein groups and 1811 genes, with 1154 protein groups and 1811 genes confidently quantified across the entire sample set. The correlation between proteomics and transcriptomics can vary from study to study, some reporting a moderate correlation [6,7,8], while some noted low correlations [5,23,24,25]. We observed a lower correlation between the proteome and transcriptome, both at the gene and pathway levels (median Pearson’s *r* = 0.43 and 0.24, respectively). The majority of the pathways that we found to be positively correlated between the proteome and transcriptome (amino acid metabolism, glycolysis, p53 pathway, adaptive immune system, DNA replication) were supported by the results from Chen et al. [5]. The significant negative correlation for the ribosomal protein *RPLP0* was corroborated by previous proteogenomic findings, in which ribosomal functions were found to be lowly correlated (an observation made not just in lung adenocarcinoma [8,25], but also in other cancer types [26]). The poor correlation for some other RNA and protein abundances, such as for members of developmental biology and the MAPK signaling pathway might be due to post-transcriptional or post-translational regulation that our study was not able to capture [27,28,29]. Regardless, the overall positive correlation between protein and gene expression suggests that the two molecular layers rather support than contradict each other.

Clustering the pROIs and tROIs in an unsupervised manner revealed that intertumoral heterogeneity is stronger than intratumoral heterogeneity, with the exception of NAT regions, which exhibited distinct molecular expression profiles. Indeed, a comparison between tumor and NAT regions resulted in a multitude of differentially expressed proteins and genes, some of which were identified at both the proteome and transcriptome level and were also supported by the CPTAC pADC dataset [6], such as the upregulation of *ALDOA* and *MUC1*, or the downregulation of *A2M* and *COL6A6*. Opposite tendencies (downregulation in tumors vs. NATs at pROI level, but upregulation in tumors vs. NATs at the tROI level) were observed for three proteins involved in the complement cascade (*C3*, *C4BPA* and *CLU*), and two proteins secreted to the ECM (*COL1A1* and *LAMB3*). Interestingly, we identified a marker, *GPX1*, which showed higher gene and protein expressions in *ALK*-rearranged pADCs compared to NATs, but not in *ALK*-negative pADCs. On the contrary, both studies by Wen et al. [30] and Tian et al. [31] identified *GPX1* as being downregulated in lung tumors compared to NATs. A previous study suggests that *GPX1* may induce cisplatin-based chemoresistance in NSCLC [32], but the exact role of this gene in lung cancer is still unclear [33]. Investigation of pathway-level differences of tumors compared to NATs mirrored known cancer hallmarks, including the impairment of glycolysis [34], unfolded protein response [35], translation [36], ECM organization [37] pathways or signaling by RTKs [38]. Importantly, except for the first two processes, these hallmarks were only detected at the protein level, highlighting the relevance of proteomics in spatial profiling studies.

The role of tumor-infiltrating immune cells and the potential routes for successful immunotherapies in NSCLC is actively studied [15,16,17,18], which is also of interest in the *ALK*-rearranged subset of lung cancers because immune-based therapies showed limited efficacy in this cancer type [39]. When examining immune-infiltration-related patterns in our data, we identified the protein *SULT1A1* being upregulated with increasing TIL %, which was confirmed only by the *ALK*-rearranged subset of pADCs in the CPTAC data [6], indicating that this protein might behave distinctly in the presence of the *ALK* oncogene. Both the upregulation of *HLA-B* and downregulation of *PTGS2* with an increasing immune score in the tROIs was supported by CPTAC data [6]. *PTGS2* (also known as *COX2*) has been associated with an immunosuppressive TME [40]. In addition, members of multiple immune-system-related processes, such as antigen processing and presentation, or neutrophil degranulation, were identified to be upregulated with an increasing immune score in the tROI data, thus confirming the agreement between the molecular signatures and the pathological evaluation. On the other hand, pROIs did not display significant pathway-level differences between regions with a varying TIL %, potentially caused by more heterogeneous immune infiltration patterns within large pROIs.

Excessive intra- and/or extra-cytoplasmic mucin can often be observed in EML4–*ALK*-positive pADCs [41,42,43]. Mucins potentially have an important, but still unelucidated role in lung cancer development [44]. It is widely known that another tissue component, the stroma, affects tumor behavior as well [45]. Our previous publication on the hereby presented proteomic data also highlighted that proteomic and glycosaminoglycan profiles of individual tumor regions are highly dependent on mucin content and less on the stromal content [19]. The GSEA analysis in our study confirmed that pROIs with higher mucin and stroma scores showed a prominent upregulation of ECM components, which again demonstrated the alignment between molecular-level findings and histologically visible ROI phenotypes. Contrasting our differential expression results with findings reported in Balbisi et al. [19] and the CPTAC data revealed that the protein products of *PRELP* and *BGN* may be promising markers for increasing the mucin and stroma content in pADC tissues, respectively.

Lung cancers with confirmed *ALK* rearrangements often contain regions that are *ALK*-negative according to a previous study [46]. This intratumoral heterogeneity of *ALK* rearrangements can potentially lead to additional molecular- and pathway-level variations, which can be captured with spatial omics analyses. By investigating which proteins and genes showed the lowest and highest variability within the tumors, we uncovered the pathways with the presumably most stable and heterogeneous activity. At both the pROI and tROI level, members of the oncogenic MAPK and PI3K–AKT–mTOR signaling exhibited low variability. Interestingly, the aberrant ALK activity in this cancer type leads to the activation of both MAPK–ERK and PI3K–AKT–mTOR pathways [47]. Focusing on pathways that contribute to intratumoral heterogeneity, both molecular layers hinted toward EMT and ECM organization-associated pathways. EMT is a crucial process in cancer, as it results in polarized epithelial cells changing their morphology to a mesenchymal phenotype, through which cells gain migratory and invasive properties [48]. This phenotypic change can mediate ALK inhibitor resistance [49]. ECM remodeling plays an essential role in the EMT process and promotes cancer metastasis [50]. Interestingly, three heat shock proteins displayed high stability at the protein expression level, but high variability at the gene expression level. Heat shock proteins are produced when cells are exposed to stressful conditions, thus the disagreement between the proteome and transcriptome might again be due to post-transcriptional regulation, which enables cells to adapt to stress in a timely manner [51].

Of note, *FN1*, a well-known EMT marker, was identified as highly variably expressed both at the protein and gene level. Moreover, our data weakly indicates that an overall higher *FN1* expression in an ALK-driven tumor tissue could be correlated with worse survival. Previous studies have shown that *FN1* has both tumor-suppressive and -promoting characteristics [52]. In line with this, it has been investigated for its role in pADC prognosis. Some noted that a lower *FN1* expression indicates a more favorable outcome [53,54], while some studies showed the downregulation of *FN1* with pADC progression [55,56] or no relationship with survival at all [57,58]. The controversies in the literature may arise from the heterogeneous expression of *FN1* within the tumor tissue, as we also demonstrated in this study. Furthermore, there is a lack of investigations regarding the specific role of *FN1* in *ALK*-rearranged pADCs.

Lastly, investigating the co-localization patterns of TME elements across tROIs provided additional insights regarding intratumoral heterogeneity of ALK-driven pADCs. Our findings indicated that the presence of naive CD8+ T cells, mast cells, myeloid DCs, neutrophils and classical monocytes was largely independent of the presence of other TME elements. The clinical significance of immune–cancer interaction patterns is highly specific to different cancer types and subtypes [59,60]. Hence, focused studies on the clinical relevance of TME localization patterns, as well as on the ALK-driven tumor cell interactions with their microenvironment are needed to further enhance our precision-medicine-based therapeutic strategies.

We acknowledge that our pilot study has limitations. Due to the small number of tumors involved with heterogeneous clinical data, our findings may not be generalizable to all ALK-driven pADCs. Furthermore, it is challenging to evaluate the individual molecular characteristics of histopathological features, as these features can be correlated with each other. Some of the observed intratumoral molecular variability might be biased from stochastic factors at the cellular level, or from phenotypical differences such as the varying levels of tumor purity. In addition, both the spatial gene expression profiling and the shotgun proteomic approach carries identification and quantification biases; the former focuses on quantifying known cancer-related genes, whereas the latter is prone to detect the proteins with a more abundant expression. Besides the unavoidable technical and biological limitations often causing milder correlations in the measured protein and transcript abundances [27,28,29], the size-differences in the studied pROIs and tROIs may also affect the aggreement between proteome- and transcriptome-level results. Large pROIs are inherent to the technique used in this study (on-tissue digestion with repeated pipetting). Although an investigation of smaller pROIs would be possible after laser capture microdissection, that workflow includes in-solution digestion which lacks the required efficiency and reproducibility [61]. Therefore, we opted to carry out the current methodology with larger pROIs, despite the size differences between pROIs and tROIs. In summary, a larger cohort of *ALK*-rearranged pADCs will be required to validate the findings presented in this paper. To strengthen the reliability of our results, however, we validated multiple findings by investigating the data of larger, previously published pADC cohorts (TCGA, CPTAC).

## 4. Materials and Methods

### 4.1. Patients and Collected Histopathological Data

The primary tumors were collected at the National Korányi Institute of Pulmonology (NKIP), Hungary, with ethical approval (2521-0/2010-1018EKU, 510/2013, 52614-4-213EKU) by the Medical Research Council of Hungary. *ALK* positivity was determined via IHC and FISH at the central pathology department. The patients were all treated with ALK inhibitors after sample collection at NKIP.

The slides were stained with hematoxylin and eosin (HE), and scanned with a Pannoramic 250b Slide scanner (3DHistech Ltd., Budapest, Hungary). Morphological areas were annotated by a board-certified pathologist with thoracic pathology experience. Immune cell (lymphocytic) infiltration was only considered intratumorally, within the area of tumor cell nests, and was assessed as a percentage semi-quantitatively in the QuPATH v0.3.0 software environment [62], and later grouped into an immune score ranging from 0 to 3. Mucin and stroma scores were given as a ratio of surface area on the tissue slide: 0: none, score 1: 1–33%, score 2: 34–66%, score 3: <67% mucin or stromal content in the given area (ROI).

### 4.2. NanoString GeoMx Profiling

Slides were baked at 65 °C for 1.5 h to be deparaffinized and then rehydrated. Antigen retrieval was performed for 20 min at 100 °C, and was digested with 1 μg/mL proteinase-K using a Leica BOND-RX (Leica Biosystems, Deer Park, Illinois, USA). After overnight hybridization with cancer transcriptome atlas (CTA) probes, the samples were washed to remove off-target probes and stained with morphology markers for two hours. The morphology markers were PanCK (488 channel, 1:500, Novus Biologicals, Centennial, CO, USA), SYTO83 (532 channel, 1:25, Invitrogen, Waltham, MA, USA), CD45 (1:100, 594 channel, Cell Signaling Technologies, Danvers, MA, USA), and CD3 (647 channel, 1:100, Origene, Rockville, MD, USA). RNA ID and UMI containing oligonucleotide tags were cleaved with UV light and collected by the GeoMx from each of the ROIs that were placed on the patient samples. Digital slides with this fluorescent staining were utilized for identification of various morphological areas and lymphocytic infiltration by the same pathologist as above. After collecting the oligo tags, NGS library preparation was performed according to Illumina protocols using Dual-Index primers to specify which ROI the tags belong to. After library purification with AMPure beads (Indianapolis, IN, USA), sequencing was performed on an Illumina NovaSeq 6000 (Illumina, San Diego, CA, USA). Fastq files were processed using the Nanostring DnD pipeline into digital count (.dcc) files, which were uploaded back into the GeoMx for data analysis.

### 4.3. Mass-Spectrometry-Based Proteomic Measurements

#### 4.3.1. On-Tissue Proteolytic Digestion and Solid-Phase Extraction Purification

The on-tissue proteolytic digestion and solid-phase extraction purification was performed on formalin-fixed, paraffin-embedded (FFPE) lung tissues. The dewaxing and rehydration step was performed by sequential washing with xylene, ethanol, ethanol-water mixtures and 10 mM ammonium bicarbonate (ABC) solution. Then, antigen retrieval was performed by heated hydrolysis (85 °C) for 30 min in sodium citrate buffer (pH = 6.0). Tissues were washed with water and dried before digestion. Tryptic digestion on the surface of the selected tissue areas (Appendix A, Appendix A) relied on a previously described protocol [61], and all incubations were performed in a humidified box placed into an environmental shaker. First, the proteins in the selected tissue areas were denatured and alkylated by adding 2 µL of a solution containing 0.1% RapiGest SF (Waters, Milford, MA, USA) + 5 mM DTT + 10% glycerol (incubation: 55 °C, 20 min). Next, 2 µL solution containing 25 mM ammonium bicarbonate + 10 mM IAA + 10% glycerol was added (incubation: in the dark, RT, 20 min). Trypsin/Lys-C mix (Promega, Madison, WI, USA) enzyme solution (2 µL) (50 ng/µL trypsin/Lys-C mix in 50 mM ABC, and 10% glycerol) was added in two cycles, followed by three cycles of 2 µL trypsin (Promega, Madison, WI, USA) enzyme solution (200 ng/µL trypsin in 50 mM ABC, and 10% glycerol). Each cycle consisted of 40 min of incubation at 37 °C. The resulting peptides were manually extracted from the tissue surface by repeated pipetting using 5 × 2 µL 10% acetic acid solution. The samples were dried in a SpeedVac and purified using a Pierce C_18_ spin tip (Thermo Fisher Scientific, Waltham, MA, USA) solid-phase extraction (SPE) method optimized in-house [63]. The resin and buffers for loading and washing were cooled to 4 °C. Sample loading was performed in 50 µL water (0.1% heptafluorobutyric acid, HFBA) and the washing step was carried out with the same buffer 2 × 100 µL). Purified peptides were eluted from the tips by adding 2 × 50 µL 70:30 *v*/*v* acetonitrile:water (0.1% trifluoroacetic acid, TFA) and 1 × 50 µL 70:30 *v*/*v* acetonitrile:water (0.1% formic acid, FA). The samples were dried down using SpeedVac.

#### 4.3.2. nanoUHPLC–MS/MS Measurements

For the nanoUHPLC–MS/MS measurements, the samples were reconstituted in 8 µL injection solvent (98:2 *v*/*v* water:acetonitrile, 0.1% FA), of which 6 µL was injected into the nanoHPLC-MS system. Dionex Ultimate 3000 RSLC nanoUHPLC coupled to a Bruker Maxis II Q-TOF (Bruker Daltonik GmbH, Bremen, Germany) via CaptiveSpray nanoBooster ionization source was used for the analysis. Trapping was performed on an Acclaim PepMap100 C18 (5 µm, 100 µm × 20mm) trap column with water containing 0.1% TFA and 0.01% HFBA. Peptides were separated on an Acquity M-Class BEH130 C18 analytical column (1.7 µm, 75 µm × 250 mm Waters, Milford, MA, USA) using gradient elution (isocratic hold at 4% for 11 min, then elevating B solvent content to 25% in 75 min, and to 40% in 15 min). Solvent A consisted of water +0.1% formic acid, Solvent B was acetonitrile +0.1% formic acid. Spectra were collected using a fixed cycle time of 2.5 s and the following scan speeds: MS spectra at 3 Hz, while CID was performed on multiply charged precursors at 16 Hz for abundant ones and at 4 Hz for low abundance ones. Internal calibration was performed by infusing sodium formate and data were automatically recalibrated using the Compass Data Analysis software 4.3 (Bruker Daltonik GmbH, Bremen, Germany).

Proteins were then identified with Byonic v4.2.10 (Swissprot human database, date of retrieval: 1 November 2020), and a focused database was created for subsequent label-free quantitation (LFQ) and intensity-based absolute quantification (iBAQ) with MaxQuant v1.6.17. The most important parameter settings are summarized in Appendix A. Further data analysis steps were applied to the resulting MaxQuant output, as described in the following paragraphs.

### 4.4. Data Analysis

Data analysis steps were performed with R v4.2.0, unless specified otherwise. The custom R scripts used for data processing and analysis are available at https://github.com/bszeitz/ALK_rearranged_pADCs_multiomics (accessed on 9 June 2023).

#### 4.4.1. Quality Control and Data Normalization

The raw NanoString data (.dcc files) were processed in the GeoMX Analysis Software v2.2.0.122. Initial QC was performed to primarily check sequencing parameters and template control counts. No transcriptomic ROIs (tROIs) were discarded. As the NanoString CTA panel has 5 probes per gene target, Biological Probe QC was run to identify any outlying probes before individual probe data were collapsed to gene-level counts. There were 9 individual outlying gene probes identified (1 each from 9 separate genes), and 1 outlying negative probe identified and removed at this stage. The resulting gene level count data are the raw dataset, and the negative probes were used to calculate the limits of quantitation for each tROI: [Geometric Mean of negative probes × (Geometric Standard Deviation of negative probes)^2^]. The raw gene counts were then normalized using third quartile (Q3) normalization before additional data analysis was performed. A normalization factor was generated for each tROI as shown: [Geometric Mean of all ROI Q3 values in the dataset/individual ROI Q3 value]. In total, 1811 genes were quantified across all tROIs. The normalized gene counts were loaded into R v4.2.0 for further processing.

For the proteomic data, normalization steps were conducted within R v4.2.0. The protein expression matrices were derived from two different quantification methods, iBAQ and LFQ, used for within-sample and between-sample comparisons, respectively. Both the iBAQ and LFQ values were log2-transformed and normalized by centering the values around the global median. A total of 2318 protein groups were quantified in at least one proteomic ROI (pROI). After retaining only 1154 and 1751 protein groups that were quantified across minimum 80% of the pROIs with the LFQ and iBAQ method, respectively, missing values were imputed with low-intensity values using the “impute.MinProb” function from the imputeLCMD R package v2.0 [20].

#### 4.4.2. Retrieval of External Proteomic and Transcriptomic Datasets

The Lung Adenocarcinoma dataset of TCGA (Firehose Legacy) was downloaded from the cBioPortal website (https://www.cbioportal.org/datasets, accessed on 8 June 2023) [64,65] on 12 April 2023. Two samples (“TCGA-50-5066-02” and “TCGA-50-5946-02”) were removed to have only one sample per patient. This resulted in a total of 522 samples in the mRNA data and 365 in the Reverse Phase Protein Array (RPPA) data. Samples with *ALK* translocation were considered for some of the analyses, for which the clinical table “data_bcr_clinical_data_patient.txt” was filtered for “ALK_TRANSLOCATION_STATUS” column = “YES”, resulting in 34 samples in the mRNA expression table, and 22 samples in the RPPA expression table.

The Clinical Proteomic Tumor Analysis Consortium (CPTAC) proteomic and transcriptomic dataset of pADCs was downloaded from the respective publication [6], accessing Appendix A. Information on the overall survival for these samples was extracted from Soltis et al., Appendix A [8]. From the 211 samples (110 tumors and 101 NATs), the samples with *ALK* translocation were selected by filtering for “ALK.fusion” column = “1”, resulting in 7 samples.

#### 4.4.3. Calculation of Single-Sample Gene Set Enrichment Scores

The normalized gene counts from NanoString and the iBAQ protein intensities (after imputation of missing values) were transformed into singscores using the “simpleScore” function from the singScore R package v1.16.0 [66]. The gene sets were obtained from the Molecular Signatures Database (MSigDB) v.7.5.1 [67]. The Hallmark [68], KEGG [69] and Reactome [70] gene sets were used in the analysis and treated as directional (i.e., gene sets that contain only upregulated genes). Gene sets where the number of overlapping genes with our data did not reach 10 were deleted. All parameters in the “simpleScore” function were left at default.

#### 4.4.4. Abundance Estimation of Tumor Microenvironment Elements

The NanoString SpatialDecon algorithm [71] was used to estimate the abundance of the immune and stromal cell types based on gene counts in individual tROIs. For this, the “spatialdecon” function from the “SpatialDecon” R package v1.6.0 and the cell profile matrix called “SafeTME” was employed. The negative control probe was used to estimate each data point’s expected background.

#### 4.4.5. Cluster Analyses

K-means clustering was performed using the “kmeans” function from the stats R package v4.2.0. For consensus clustering of the full pROI and tROI data, the ConsensusClusterPlus R package [72] v1.60.0 and its function “ConsensusClusterPlus” was used. The basis of clustering was the pROIs’ protein LFQ values and tROIs’ gene counts. Both expression matrices were Z-score-normalized, and then resampled 1000 times using the bootstrap method with a probability of 0.8 for selecting any sample and any protein/gene. The bootstrap sample datasets were clustered using the partitioning around medoids method with Pearson distance and complete linkage. The range of two to ten clusters were explored, and the best number of clusters (K) was selected based on the visual inspection of ConsensusClusterPlus outputs.

#### 4.4.6. Correlation Analyses

To assess the similarity of the two omic datasets, the gene count matrix and singscore matrix of the smaller tROIs from the same pROI were averaged. These average gene counts and singscores of tROIs were correlated with pROI LFQ values (filtered and imputed) and singscores, respectively, using Pearson correlation, by applying the “cor.test” function from the correlation R package v0.8.2 [73]. The *p*-values were corrected using the Benjamini–Hochberg (BH) method and adj. *p* < 0.05 was considered significant.

#### 4.4.7. Differential Expression Analyses

Differential expression analyses using gene counts and protein LFQ values were performed using the R package glmmSeq v0.5.5 [74]. The “lmmSeq” function was used, building linear mixed effects models in which the patient was always included as a random effect in the model.

For tumor vs. NAT comparisons, only patient tumors containing both tissue types were considered (five patients for the tROIs and three patients for the pROIs). Only those proteins/genes were regarded as significant which showed a minimum of 1.5-fold change (FC) difference and BH-adj. *p* < 0.05.

To identify features that are correlated with the extent of immune cell infiltration in the tumors, TIL % as a continuous independent variable for the pROIs (ranging between 5 and 40 %) and immune score as categorical independent variable for the tROIs (ranging between 1 and 3) were used. In case of pROI results, the only proteins regarded as significant were those for which BH-adj. *p* < 0.05. Regarding tROI results, the only genes regarded as significant were those which showed an upregulation tendency from immune score 1–2–3, with a minimum of 1.5 FC difference and BH-adj. *p* < 0.05.

Mucin and stroma score comparisons were only checked in the proteomic data. Prior to the mucin score analysis, the NAT regions and the only region with mucin score = 1 were excluded. The only proteins regarded as significant were those which showed an upregulation tendency from mucin score 0–2–3, with a minimum of 1.5 FC difference and BH-adj. *p* < 0.05. Prior to the analysis of stroma score differences, the NAT regions and the only region with stroma score = 0 were excluded. The only proteins regarded as significant were those which showed an upregulation tendency from stroma score 1–2–3, with a minimum of 1.5 FC difference and BH-adj. *p* < 0.05.

#### 4.4.8. Cox Regression Analyses

Cox regression analyses were performed using the “coxph” function from the survival R package v3.4.0. For the pROI and tROI data, the mean ROI values (pROIs: LFQ intensities; tROIs: gene counts) within each patient were taken prior to the analysis to have only one expression value per patient. NAT regions were excluded from this calculation. The Cox regression models were adjusted for different baseline hazards of patients receiving Crizotinib and Alectinib (“strata(drug)”). For external datasets (TCGA, CPTAC), the “strata(stage)” expression in the Cox model was used to adjust for the different baseline hazards of patients.

#### 4.4.9. Selection of Proteins and Genes with Stable and Variable Expression within Tumors

To address the variability of their expression, the CV was calculated for each protein and gene individually within each patient’s tumor. The ROIs of NATs were excluded prior to CV calculation so that the analysis was not driven by tumor vs. NAT differences, thus proteomic data of Case 2 was not included in the analysis as it contained only one tumor pROI. Proteins and genes with stable expression were selected based on whether their CV values were among the bottom 20% within at least four cases. The proteins and genes with variable expression were chosen according to their CV values being among the top 20% within at least four cases.

#### 4.4.10. Pathway Analyses

To perform one-tailed Fisher’s exact test (i.e., overrepresentation analysis, ORA) of the gene sets from Hallmark, KEGG, Reactome databases, the “fora” function of the “fgsea” R package v1.22.0 [75] was used. The background gene list (“universe”) was set according to the conducted analysis. When the pathway enrichment for genes positively correlating between pROIs and tROIs were assessed, only the commonly quantified 162 genes were used; when any list of proteins from the pROI data was investigated, then all proteins quantified in at least one pROI were used (*n* = 2318); and when any gene list from tROIs was checked, then all genes quantified across the tROIs were used (*n* = 1811). The minimum and maximum size of the gene sets were set to 1 and infinite, respectively.

Differential expression analyses were followed by pre-ranked GSEA, in which the Hallmark, KEGG, Reactome gene sets were tested. The proteins/genes were ranked based on the FC/coefficient multiplied by the −log10 *p*-value. In the case of multiple-group comparisons when multiple FCs/coefficients were present, the mean FC/coefficient was used for pre-ranked GSEA. The pre-ranked GSEA was performed using the “GSEA” function from the clusterProfiler R package v4.4.4 [76]. The default settings were used, except that pvalueCutoff was set to 1. The BH method was used for *p*-value adjustment. For visualization purposes, representative pathways from all significant pathways were curated by the authors.

#### 4.4.11. Visualizations

Figures were drawn using the R packages ggbiplot v0.55, ComplexHeatmap v2.1.13 and ggplot2 v3.4.1.

## 5. Conclusions

Advances in omics profiling technologies, such as next-generation sequencing or mass-spectrometry-based proteomics, have enabled a more comprehensive molecular profiling of lung tumors, leading to the identification of novel molecular alterations and the development of new targeted therapies. This pilot study, to our knowledge, is the first to address the inter- and intratumoral heterogeneity of *ALK*-rearranged pADCs at both the proteome and transcriptome levels. The gene and protein expression patterns within the tumors and across histologically distinct regions may have future implications for targeted therapy design. By uncovering the spatially resolved biology of *ALK*-rearranged tumors, we hope this work will enhance our biological understanding and improve therapeutic efforts for this cancer type.

## Figures and Tables

**Figure 1 ijms-24-11369-f001:**
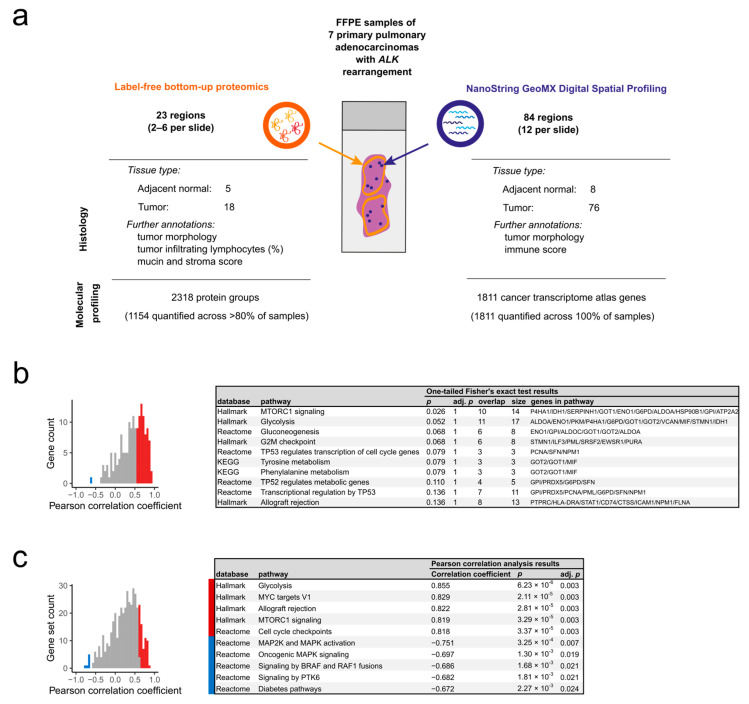
Spatial characterization of the proteome and transcriptome in seven *ALK*-rearranged pADCs. (**a**) Schematic overview of the study. (**b**) Histogram of gene-wise Pearson correlation coefficients calculated between the pROIs and tROIs (**left**) and the enriched pathways (one-tailed Fisher’s exact test, *p* < 0.15) for the significantly positively correlated genes (**right**). (**c**) Histogram of Pearson correlation coefficients calculated between the pROIs and tROIs at the singscore level (**left**), and the top significantly correlating gene sets (**right**).

**Figure 2 ijms-24-11369-f002:**
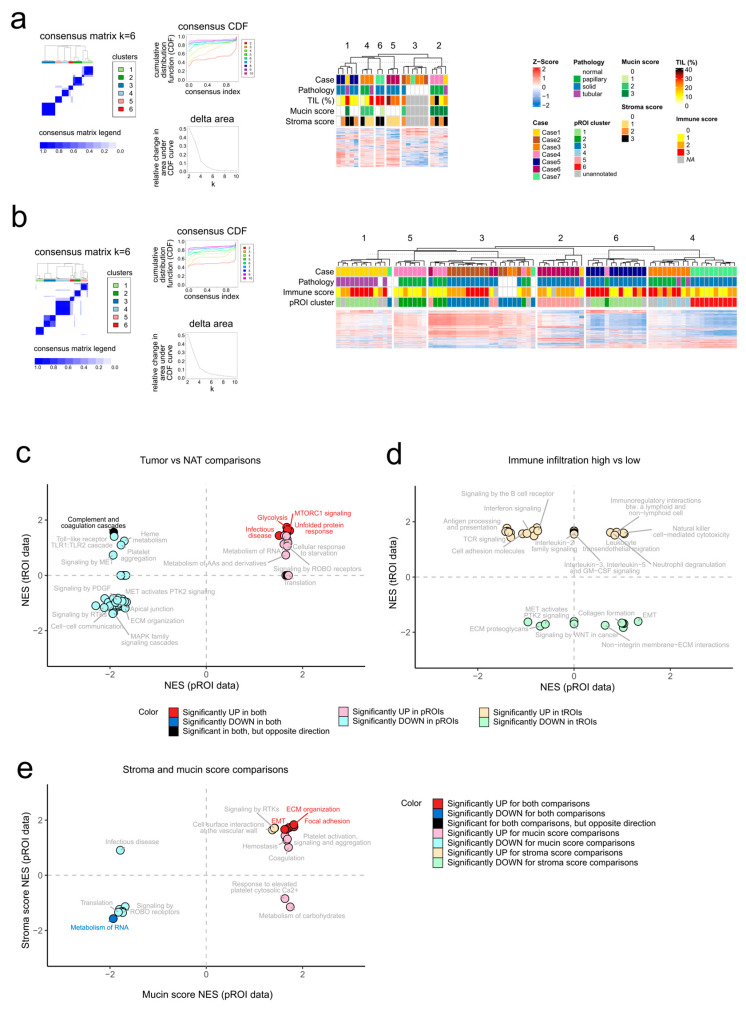
Investigation of histopathological features at the multi-omic level. (**a**) Consensus clustering outputs for pROIs based on the normalized protein LFQ values (**left**) and heatmap showing the pROI clusters (**right**). (**b**) Consensus clustering outputs for tROIs based on the normalized gene counts (**left**) and heatmap showing the tROI clusters (**right**). (**c**) Normalized enrichment scores (NES) from pre-ranked GSEA for tumor vs. NAT comparisons in pROI (x-axis) and tROI (y-axis) data. (**d**) NES from pre-ranked GSEA for immune infiltration high vs. low comparison in pROI (x-axis) and tROI (y-axis) data. (**e**) NES from pre-ranked GSEA for the mucin high vs. low (x-axis) and stroma high vs. low (y-axis) comparisons in the pROI data.

**Figure 3 ijms-24-11369-f003:**
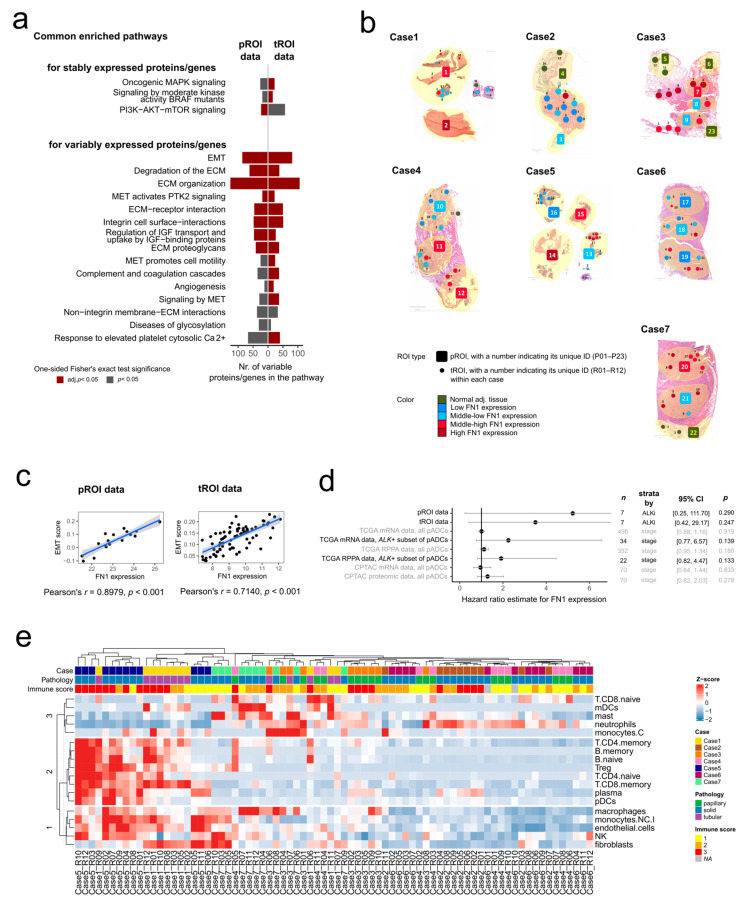
Observed intratumoral homogeneity and heterogeneity in *ALK*-rearranged pADCs. (**a**) Enriched gene sets for the proteins and genes showing low or high variability within a minimum of four tumors, supported by both pROI and tROI data. (**b**) Tumor ROIs showing a heterogeneous expression of *FN1* at both protein and gene level. The *FN1* categories were defined by clustering ROIs based on *FN1* expression (Euclidean distance and complete linkage). (**c**) *FN1* expression vs. EMT singscore across the pROIs (**left**) and tROIs (**right**). (**d**) The *FN1* protein/gene expression association with survival both in this study’s cohort and in the TCGA/CPTAC pADC cohorts. The hazard ratios with 95% confidence intervals (CIs) and Cox regression *p*-values are indicated on the right. (**e**) Unsupervised clustering of the estimated abundance of TME elements in the tROIs.

**Table 1 ijms-24-11369-t001:** Summary of clinical and histopathological data for each patient. ALKi, ALK inhibitor; avr., average; nr., number; NA, not available; NAT, normal adjacent tissue; OS, overall survival; pROI, proteomic region of interest; TIL, tumor infiltrating lymphocyte; tROI, transcriptomic region of interest; yrs, years.

Information		Case 1	Case 2	Case 3	Case 4	Case 5	Case 6	Case 7
Clinical data	Sex	male	male	male	female	female	female	male
	Age at diagnosis (yrs)	53.6	43.7	68.9	32.8	68.5	64.2	66.7
	Stage on presentation	NA	NA	NA	3	NA	4	3
	Administered ALKi	Crizotinib	Crizotinib	Crizotinib	Crizotinib	Crizotinib	Alectinib	Alectinib
	Alive	no	yes	no	yes	yes	yes	yes
	OS (yrs)	2.2	6.6	4.1	13.8	7.3	4.9	6.7
Proteomic data	Nr. of pROIs	2	2	6	3	4	3	3
	Morphology of pROIs (nr.)	tubular (2)	NAT (1), solid (1)	NAT (3), papillary (2), tubular (1)	papillary (3)	solid (4)	solid (3)	NAT (1), solid (2)
	Avr. TIL (%)	25.00	25.00	15.00	23.33	8.75	28.33	30.00
	Avr. mucin score	3.00	2.00	2.00	3.00	0.00	0.00	0.50
	Avr. stroma score	3.00	2.00	1.67	2.33	1.75	1.00	3.00
Transcriptomic data	Nr. of tROIs	12	12	12	12	12	12	12
	Morphology of tROIs (nr.)	NAT (1), tubular (11)	NAT (2), solid (10)	NAT (2), papillary (8), tubular (2)	NAT (1), papillary (11)	solid (12)	solid (12)	NAT (2), solid (10)
	Avr. immune score	2.25	2.00	2.00	1.42	2.25	1.60	1.17

## Data Availability

The raw data of the proteomics measurements are available in the MassIVE repository at the https://doi.org/10.25345/C5319S63V link, and can be downloaded via FTP (ftp://massive.ucsd.edu/MSV000089286/) (accessed on 9 June 2023). The clinical data and ROI annotations, as well as the normalized gene counts from NanoString GeoMx profiling and the normalized LFQ/iBAQ protein intensities, are available in Appendix A. The publicly archived datasets from TCGA and CPTAC used in this study are available at https://www.cbioportal.org/ (accessed on 8 June 2023) and in publication [6]. Data analysis scripts can be obtained at https://github.com/bszeitz/ALK_rearranged_pADCs_multiomics (accessed on 9 June 2023) to reproduce the findings presented in this paper.

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
