# Peer review of "Spatially Resolved Proteomic and Transcriptomic Profiling of Anaplastic Lymphoma Kinase-Rearranged Pulmonary Adenocarcinomas Reveals Key Players in Inter- and Intratumoral Heterogeneity"

_ijms, 2023, doi:10.3390/ijms241411369_

Round 1

Reviewer 1 Report

Comments and Suggestions for Authors

According to the presented data, the authors are investigating pulmonary adenocarcinomas using multi-omics techniques. Proteomics and genomics data along with clinical and demographical data were used to get spatial information on the components of the tumor microenvironment. The statistical analyses along with pathway analysis could reveal further information on intratumoral heterogeneity. The workflow applied is interesting and can provide a useful guide for further similar studies. The comparison of the obtained data with the CPTAC dataset is another added value.

The study is well-designed and has high clinical relevance. The presentation of the data is clear; the figures are well structured. I have some really minor points which need to be addressed:

In the discussion it is mentioned that „we achieved the confident quantification of 1154 proteins” Isn’t it 1164? In the results part it was indicated that missing values were inputed for 1154 proteins. Please clarify the situation.

In line 130 the „2318 proteins in at least one ROI” is a bit confusing. Please revise this part.

Line 212: In the sentence „Moreover, proteomics indicated that pathways such as translation and metabolism of RNA, glucose and amino acid were more active in tumors compared to NATs.” I think proteomics results do not show activity. A higher abundance of proteins and enriched pathways do not give any information on enzyme activity. As far as many of the proteins involved in these pathways are enzymes, please rephrase this sentence omitting the activity.

In the study, the authors concentrated on the overlaps between the transcriptomic and proteomic data. What about the differences? Are they meaningful?

It is in the results, but please emphasize a bit more the unique data got with proteomics results. Proteins are the ones who really perform the biological functions, so please highlight the value of the proteomics results.

Reviewer 2 Report

Comments and Suggestions for Authors

1.     Anaplastic lymphoma kinase rearrangements occur in small portion of patients with pulmonary adenocarcinoma. This ALK gene arrangement causes constitutive activation of ALK kinase and subsequent ALK driven tumor formation. Despite the ALK kinase inhibitors such as Crizotinib and Alectinib improve the patients’ survival, majority of patients develop acquired resistance which compromises the durable response of these drugs. In the present study, by analyzing spatial proteomic and transcriptomic profiles in ALK positive tumors, authors were revealed the targets of heterogeneity in tumors which may be useful for future therapeutic approaches.

2.     In present study, authors have used advanced methods such as nanostring Geomax and mass spectrometry to identify the spatially expressed genes and proteins to determine tumor heterogeneity. However, authors have only used treatment naïve pADC tumors to investigate intra and inter heterogeneity and found out that inter tumoral heterogeneity is pronounced than intra tumoral heterogeneity. If the authors compare the tumor heterogeneity between the tumors that respond to ALK inhibitors, tumors with primary resistance, and acquired resistance that would be better study that will give the information about molecular targets or dysregulated pathways that contributing to primary and acquired resistance.   

3.     Since pADC tumors with ALK rearrangements are resistance to immunotherapy. If the authors have shown difference in tumor heterogeneity between the tumors that respond and resistance to immunotherapy that would have become more impact research.

4.     Authors were used smaller ROI for transcriptome and larger ROI for proteome spatial profiling. What is the rationale behind different sizes of ROI for transcriptome and proteome? Size of ROI should be same for both transcriptome and proteome to compare the expression levels of genes and proteins. This may be the reason of modest correlation of transcriptome and proteome.

Round 2

Reviewer 2 Report

Comments and Suggestions for Authors

Authors are addressed my comments clearly with reasonable explanation. I would agree this manuscript for publication.